# The impact of girl child marriage on the completion of the first cycle of secondary education in Zimbabwe: A propensity score analysis

**Annah V. Bengesai**[1]*, **Lateef B. Amusa**[2], **Felix Makonye**[1]

**1** College of Law and Management Studies, University of KwaZulu-Natal, Durban, South Africa,
**2** Department of Statistics, University of Ilorin, Ilorin, Nigeria

\* bengesai@ukzn.ac.za

## Abstract

### Background

The association between girl child marriage and education is widely acknowledged; however, there is no large body of demographic studies from Zimbabwe that have addressed this aspect. This study aimed to examine the extent to which child marriage affects one academic milestone, i.e. completion of the Ordinary Level, the first cycle of high school, which is also the most critical indicator of educational achievement in Zimbabwe.

### Methods

We used the 2015 Zimbabwe Demographic and Health Survey and extracted 2380 cases of ever-married women aged between 20–29 years. We applied a propensity score-based method, which allowed us to mimic a hypothetical experiment and estimate outcomes between treated and untreated subjects.

### Results

Our results suggest that child age at first marriage is concentrated between the ages of 15–22, with the typical age at first marriage being 18 years. Both logistic regression and PSM models revealed that early marriage decreased the chances of completing the first cycle of high school. Regression adjustment produced an estimate of prevalence ratio (PR) of 0.446 (95% CI: 0.374–0.532), while PSM resulted in an estimate (PR = 0.381; 95% CI: 0.298–0.488).

### Conclusion

These results have implications for Zimbabwe's development policy and suggest that girl-child marriage is a significant barrier to educational attainment. If not addressed, the country will most likely fail to meet sustainable development Goal 4.2 and 5.3. Social change interventions that target adults and counter beliefs about adolescent sexuality and prepubescent

**Data Availability Statement:** The data underlying the results presented in the study are available from theDemographic and Health Survey (DHS)

programme and can be accessed from https://dhsprogram.com/Data/.

**Funding:** The authors received no specific funding for this work.

**Competing interests:** The authors have declared that no competing interests exist.

marriage should be put in place. Moreover, interventions that keep teenage girls in school beyond the first cycle of high school should be prioritised.

## Introduction

Although many countries globally have ratified the Convention of the Rights of the Child's [1] definition that any human being below the age of 18 is a child, girl child marriage is still a common practice in many parts of the world, especially in South Asia and sub-Saharan Africa [2]. Globally, an estimated 39 000 child marriages occur every day [3], and if these trends persist, approximately150 million girls will be married off by 2030, many against their will [4]. Statistics from sub-Saharan Africa indicate that the proportion of women who marry before their 18[th] birthday ranges from 19% in Namibia to more than 70% in Chad, Mali, and Niger [5]. It is also estimated that nearly 50% of the world's child brides live in South Asia, while India alone accounts for about 30% of the global total [2]. Even in developed countries such as the United States of America, the statistics suggest that approximately 78400 children have been married between 2010 and 2014, with immigrant children being more likely to be married than their peers born in the United States [6].

Like in most other parts of Africa, Zimbabwe also has a high prevalence of girl child marriage throughout its 10 provinces, despite the awareness campaigns held in the past few decades [7]. For instance, recent data from the Multiple Indicators Survey [8] estimates that the country's child marriage rate is 33%, slightly higher than the global average of 29%. Consequently, Zimbabwe is classified among the 41 countries globally with an alarming rate of child marriages [9, 10].

These statistics on child marriage are not only alarming but also suggest that the practice continues unabated. This is especially concerning because many such marriages often go unreported or unregistered, especially in rural areas [11]. Therefore, the exact scope of the problem might still be unknown, suggesting that child marriage might be a hidden and unaddressed problem [12].

Several reasons, most of which are steeped in the family institution, culture and the toxic combination between poverty and gender discrimination, have facilitated the continual exploitation of the girl child [12–15]. Some scholars have noted that poor parents are often compelled to marry off their daughters when faced with austerity, thus reducing their expenses as they will have one less person to feed, clothe and educate [16]. In some communities, the onset of menarche is considered the threshold for adulthood and a sign of marriage readiness [14, 17]. Also, some believe that getting a girl married early might have a protective factor against early sexual debut, loss of virginity, factors which if not protected, will affect the family's honour [18].

In Zimbabwe, there are three main drivers of child marriage: cultural and (forced) marriage practices; religion; and, lack of policy enforcement. Regarding the latter, Hallfors et al. [19] note that while Zimbabwe's constitution prohibits child marriages, the enforcement part is lacking. Forced marriage practices have prevailed despite the country being a signatory to the African Charter on the Rights and Welfare of the Child [20], which places an obligation on member states to end harmful practices such as child marriage, put in place effective actions, including monitoring the progress towards eradicating such practices [21]. In other words, prohibiting child marriage by law alone has had little effect in eliminating the practice [19, 22], perhaps due to "countervailing norms, or widespread exceptions" [23], some of which are discussed below.

Outside of policy, retrogressive marriage practices such as *kuripa ngozi* (virgin pledging), *kutizira* (unplanned pregnant marriages) and *kuzvarira* (pledged marriages) are also continually used to justify marrying off young girls in Zimbabwe [24]. In *kuripa ngozi*, *young* girls are married off to appease an avenging spirit, while *kuzvarira* is often a survival tactic where low-income families negotiate with wealthy families to marry off their daughters at a younger age in exchange for grains, cows or money. *Kutizira* is a practice where if a girl gets pregnant out of wedlock, she is expected to elope to her boyfriend or the person responsible for the pregnancy. This is done to mitigate the shame of premarital sex and childbearing out of wedlock while preserving the family honour.

The prevalence of child marriage in Zimbabwe has also been linked to religion. While the practice is not peculiar to one religious group, Chamisa et al. [25] note that girl child marriages are more prevalent among the apostolic religion adherents, especially the Johane Marange and Johane Masowe groups. These religious groups exploit young girls through their religious teachings and practices. For instance, it is common for young girls to be forced into marrying older men under the pretext that church leaders have been directed by the 'holy spirit' [19]. Those who fail to adhere to these 'prophecies' are threatened and cursed, forcing them to do certain things, even against their will. This practice has also endured due to the strong political ties between the Government of Zimbabwe and the Apostolic Faith groups, which commands a considerable following estimated to be about 34% of the country's population [26]. As a result, Zimbabwean politicians have targeted this religious sect, using its pulpit to garner electoral support [19]. Critics have argued that these religious-political ties perhaps explain why policymakers tend to overlook child marriage issues among the Apostolic Faith sect as they fear losing a significant proportion of the electorate [19, 25].

Whatever the reasons for marrying off young girls, child marriage is a human right violation, with far-reaching consequences. Child marriage robs young girls of their childhood and is a significant setback for development [12]. It hinders their full participation in society as well as efforts to achieve gender equality broadly [7, 21]. Early marriage also confers risks to sexually transmitted diseases (STIs), including HIV and AIDS, early childbearing and the associated health hazards [27]. There is also evidence that child brides are more likely to become domestic violence victims and grow up feeling disempowered [15].

Undoubtedly, the demographic and health costs of early marriage are high, and governments across the globe cannot afford to allow this practice to continue. It is not surprising that several national and global initiatives have been put in place to find solutions to the problem. Child marriage is now a major policy issue and included in sustainable development goal no 5 [13, 28]. Global organisations and development partners such as UNICEF, UNFPA and Girls not Brides have also been working with many governments to improve the status of the girl child. Specific to Zimbabwe, platforms such as Girl Child Network and Childline have been used to raise awareness of the extent of the problem; however, progress remains slow and uneven [25].

Among the different solutions that have been proposed, education is considered the most important protective factor against girl child marriage [12, 13]. This is not surprising given that the returns to education, particularly secondary level education, are well documented [12, 13, 29]. Scholars have argued that lack of education curtails young girls' full realisation of their rights, including the right to object to forced marriages and limiting their livelihood options [30]. However, the relationship between child marriage and education is not as straightforward, as it is influenced by several interconnected processes. For instance, some young girls may be forced to drop out of school due to poverty or academic ability, and, in a patriarchal society, marrying them off might be seen as the next rational step. At the same time, girls who are married early might fail to continue with their education due to different factors such as

the financial situation, caring for a baby or simply because the husband might not allow them to [13, 31]. Put differently, child marriage could be a driver or a consequence of low educational attainment. To tease out the possible association with educational attainment, there is a need for methods that can isolate the true effect of child marriage from any confounding. However, there is not a large body of demographic studies that have addressed this aspect [13, 31].

To put Zimbabwe into context, studies on child marriage, in general, are limited. Where these have been done, the focus has been on determinants or prevalence [7, 25, 32, 33], legal and development focused efforts to end the practice [9, 34] or the association with reproductive health [19, 35]. Moreover, most of these studies on child marriage have been small scale, based on -non-probability sampling and limited to geographic areas, with population-based studies [32] being undertaken infrequently. Notably lacking is empirical research that has focused on the association between child marriage and educational outcomes, particularly secondary level schooling, which is the focus of the present study. Therefore, this study aims to address these gaps by examining the impact of early marriage on completing the first cycle of secondary school. We also draw on a propensity score-based approach that allows us to measure the effect of early marriage on educational attainment.

## Materials and methods

### Data

The data used in this study came from the 2015 Zimbabwe Demographic and Health Survey (ZDHS), a cross-sectional nationally representative study of health and demographic indicators of women aged 15–49. Mindful that our focus was on completion of the first cycle of secondary school (11[th] year of schooling), which cannot be achieved before the age of 17 years if one is on time, we restricted our analysis to 2380 ever married or partnered women who were aged 20–29. Thus, we gave at least three more years for the youngest cohort to complete this level of schooling. We also excluded women who were not in their first union as they would not have provided information related to their first spouse. This is particularly important in cases where the first union was before the woman's 18[th] birthday.

The 2015 ZDHS, was undertaken by the Zimbabwe National Statistics Agency (ZIMSTAT) in collaboration with the Ministry of Health and Child Care (MoHCC) and the United Nations Population Fund (UNFPA). Data were collected from approximately 11 000 households which were sampled using a two-stage cluster sampling design with two strata (urban and rural) for each province. At the first stage, 400 enumeration areas delineated by the 2012 Zimbabwe Population Census sampling frame were selected, of which 166 were urban areas and 234 rural areas. The second stage involved the selection of individuals at the household level.

### Measures

The outcome variable was the completion of lower secondary school, while early marriage was the treatment variable. We present the descriptions of the variables as used in this study below:

### Outcome

Lower secondary school completion (yes = 1, no = 0), indicated whether an individual had completed the first cycle of high school or not. In the ZDHS, respondents were asked questions about the highest grade or years of schooling they had completed by the survey date. We considered individuals who had completed 11 years of schooling as having completed lower secondary education. Zimbabwe has a 7-4-2 basic education structure consisting of seven years of

primary, followed by two cycles of secondary level education; four years of General Certificate of Education, also called Ordinary level and two years of General Certificate of Education at the Advanced Level [36]. The Ordinary level, which is the 11<sup>th</sup> year of schooling, is generally used to determine student achievement, progression to either A-level, polytechnic or teachers' colleges, as well as employment status. While learners who want to progress to university might opt to pursue the Advanced Level Education, the majority, due to several factors such as quality of pass, lack of fees or career choice, often end at the Ordinary level. Thus, completion of this level is the most critical indicator of educational achievement, such that a person who holds the Ordinary level certificate is referred to informally as having 'finished school'. In consequence, most young people 'finish school' as early as 17 years old. Given the fragile economic landscape in Zimbabwe, characterised by high unemployment, closure of companies, retrenchments and poverty, marriage often becomes the next best option for young people who are not in school.

## Treatment variable

Our main explanatory or treatment variable was *early marriage*, a binary indicator based on age at marriage (<18 years, and ≥18 years).

## Other variables of interest

We also included the following factors that were found in the previous research to be associated with early marriage and developmental outcomes such as education and coded them as follows:

## Union characteristics

- Spousal age difference has been identified as a proxy for female autonomy in reproductive health research. Using data on the woman and her partner's actual age, which was captured as a continuous variable in the ZDHS, we created the spousal age difference variable by subtracting the wife's age from that of her partner.

- Polygamous marriage: This was a dummy variable to distinguish between (0 = monogamous marriage and, 1 = polygamous marriage).

- Husband's education: 1 = primary level education or lower; 2 = secondary level education, 3 = higher education.

**Sexual debut.** Early sexual debut is often associated with teenage pregnancy, and both factors have a reciprocal effect on both early marriage and educational attainment [12]. At the same time, early marriage can also determine sexual debut for many young girls, by extension, placing the latter on the causal pathway between child marriage and educational attainment. Mindful of this, we opted to distinguish between women who had their sexual debut before marriage and those whose age at first sex was after marriage. Hence, we use this variable as one of the controls and coded this as a binary variable to; (1 = before marriage; 2 = after marriage).

**Sex of the household head.** We also included the sex of the household head (0 = male; 1 = female) as a control. Female-headed households are increasingly becoming common in Zimbabwe, and research evidence suggests a higher incidence of poverty in these households relative to male-headed ones [37]. As such, it is often suspected that female-headed households have limited resources to invest in their children's education, with girls being the most affected.

## Sociodemographic controls

Given the literature which suggests that a nexus between religious beliefs and child marriage [25] as well as the prevalence of the latter in rural areas [24], we also included the following covariates: the place of residence (1 = rural, 2 = urban); and religion (1 = christian, 2 = apostolic, 3 = other).

## Analytical method

We first explored the associations between the treatment variable (a binary indicator of early marriage) and potential confounders. Since analyses were survey-weighted, we used the Rao-Scott $X^2$ test for the categorical variables and the weighted t-test for the continuous variables (Table 1).

Given that our primary objective was to quantify the effect of early marriage, we opted for a methodology that controls for confounding factors. Therefore, we used propensity score matching (PSM) to estimate treatment effects and compared the results with those obtained from the conventional regression adjustment and the unadjusted estimate of the treatment effect. Though some methods of using the propensity score (PS) have been described in the statistical literature [38], we favoured PSM in this study due to its convenience in implementation and computation.

In this setting, we defined the PS as the conditional probability that a participant had an early marriage, conditional on the covariates. For the propensity score analysis, two steps were involved. First, we performed a binary logistic regression model of main effects by regressing the treatment variable (a binary indicator of early marriage) on all the covariates identified as potential confounders in Table 1 to generate propensity scores. We adopted the strategy of

**Table 1. Descriptive statistics (% and means) for child marriage, and associations with selected variables of married women aged 20–29, 2015 ZDHS, n = 2380.**

| Variable | Early marriage | Late Marriage | Raw data | | |
|---|---|---|---|---|---|
| | N = 821 (36.8%) | N = 1559 (63.2%) | P-value | ASMD | Matched ASMD |
| *Place of residence* | | | <0.001 | | |
| Rural | 76.2 | 55.4 | | 0.528 | 0.018 |
| *Religion* | | | <0.001 | | |
| Christian | 35.5 | 54.9 | | 0.441 | 0.021 |
| Apostolic | 54.8 | 39.9 | | 0.346 | 0.009 |
| Others | 9.7 | 5.2 | | - | - |
| *Husband education (years)* | | | <0.001 | | |
| 0–10 | 31.3 | 19.3 | | 0.215 | 0.009 |
| 11+ | 23.9 | 33.8 | | 0.04 | 0.011 |
| Missing | 44.8 | 46.8 | | - | - |
| *Type of marital union* | | | <0.001 | | |
| Monogamy | 73.2 | 82.1 | | 0.278 | 0.038 |
| Polygamy | 16.2 | 13.8 | | 0.276 | 0.047 |
| missing | 10.6 | 4.1 | | - | - |
| *Age at first sex* | | | <0.001 | | |
| Before marriage | 17.3 | 41.7 | | 0.586 | 0.002 |
| *Sex of household head* | | | 0.134 | | |
| Female | 34.0 | 37.6 | | 0.125 | 0.008 |
| *Spousal age difference* [b] | 7.3±5.1 | 5.7±4.8 | <0.001 | 0.344 | 0.028 |
| *Age of respondent (years)* [b] | 24.2±2.7 | 25.1±2.9 | <0.001 | 0.318 | 0.058 |

Note: Reported percentages are population-weighted. We only report p-values for the original data and ASMDs for the matched data [43]. ASMD means absolute standardised mean difference and is a numeric value calculated for every covariate. An ASMD of <0.1 is generally taken to indicate a negligible difference between the treatment and the control group for that covariate [45, 46].

[b] mean ± standard deviation.

including as many covariates as possible based on previous research and scientific understanding [39, 40]. Second, we used the estimated propensity scores to create a matched sample based on the single nearest neighbour algorithm [41–43]. We implemented a 1–1 matching with replacement and without caliper.

We assessed the estimated propensity scores for sufficient overlap in the distribution of the treated and control groups (Fig 2). The PS adjustment objective is to create a matched sample in which the distribution of the covariates is the same between the treatment and control groups. We thus verified that covariate balance had been induced in the matched samples by examining their pre-matched and post-matched ASMDs [44]. While there is no consensus cut-off, ASMD values >10% may indicate covariate imbalance [45, 46].

We then determined the effect of early marriage by estimating the prevalence of the outcome in treatment and control respondents separately in the matched sample. The prevalence estimation incorporated the survey weights to obtain population-level estimates [47]. The treatment effect was quantified as the prevalence ratio (PR) of early marriage respondents who completed the Ordinary Level of secondary schooling to the late marriage respondents who also completed the same level. For the regression adjustment, the traditional multivariable logistic regression was used to calculate the adjusted PR. However, the survey weights were accounted for in the analysis to account for the survey design. Further, where necessary, variance estimations accounted for the design of the survey. While we acknowledge the high likelihood of inducing bias, the missing indicator method [48] was applied to missing values on husband education to increase the sample size for the propensity score and multivariable models. We carried out all analyses in Stata v14. Where necessary, we reported two-sided p-values and p-values < 0.05 were considered statistically significant.

## Results and discussion

Fig 1 shows the distribution of the sample by age at first marriage. We find that age at first marriage is concentrated between the ages of 15–22. The typical age at first marriage is 18, followed by 17 and 19 in that order.

Table 1 shows the weighted descriptive statistics of the association between different variables and early marriage. We observed that the married respondents in their adolescence

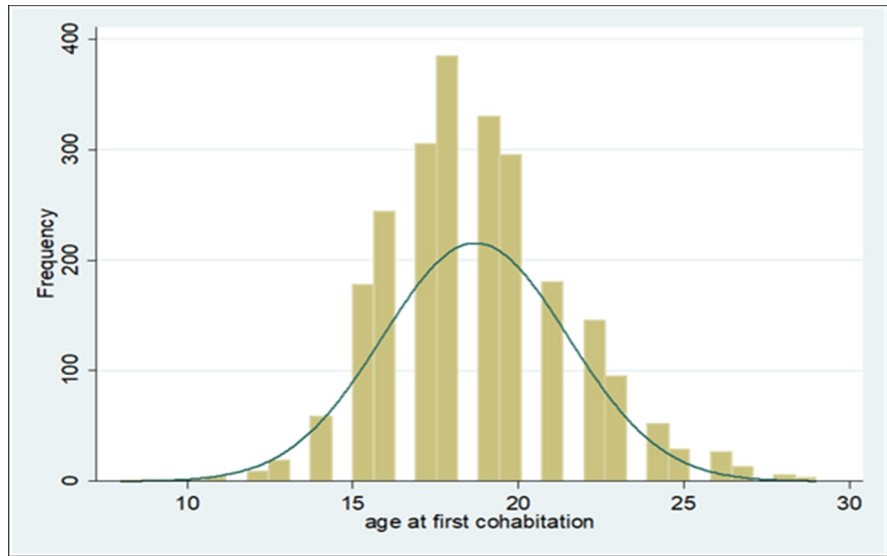

**Fig 1. Sample distribution of age at first cohabitation.**

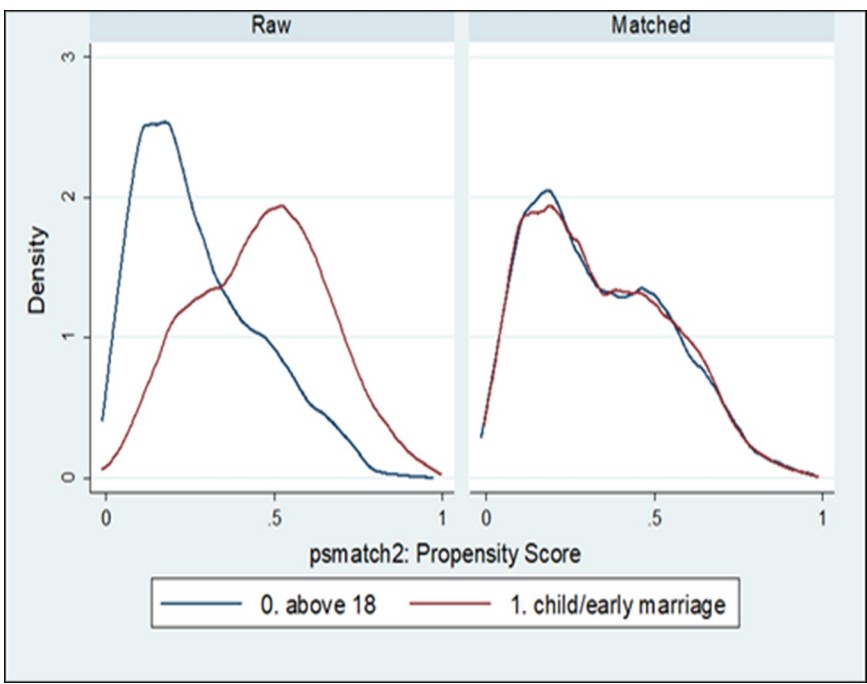

**Fig 2. The distributions of the estimated propensity scores for the early and late marriage participants before and after matching.**

accounted for 36.8% of the total sample, while 63.2% had married after their 18th birthday. Table 1 shows that all our covariates (column 4) were significantly associated with the treatment variable, girl child marriage. There were statistically significant differences (p < 0.05) in all the confounders between the two groups, implying systematic differences in the risk factors between the two groups of interest.

Fig 2 is a density plot that shows the propensity scores distribution, both pre-matching and post-matching, for the two groups of age at marriage. As shown in the figure, compared to pre-matching, the propensity scores distribution was almost identical for the two groups after matching on the propensity score. After applying PSM, all the covariates were balanced, and there were no statistically significant differences (p > 0.05) between the two groups (results not shown but available on request). The ASMD values (Table 1, columns 5 and 6) most of the covariates were also substantially under the 0.10 threshold [43].

We do not present the results from the covariates' estimates obtained in the first logistic regression (although these are available upon request). Both the unadjusted and all the adjusted estimates suggested that early marriage significantly (p < 0.05) associated with high school completion (Table 2). However, all the adjusted estimates increased the PR from the

**Table 2. Association of child marriage and completion of lower secondary school.**

|  | Unadjusted | Regression-adjusted | PSM |
|---|---|---|---|
| Early marriage prevalence (%) | 0.199 | 0.236 | 0.173 |
| Late marriage prevalence (%) | 0.609 | 0.529 | 0.454 |
| PR (95% CI) | 0.328 (0.285, 0.378) | 0.446 (0.374, 0.532) | 0.381 (0.298, 0.488) |
| P-value | <0.0001 | <0.0001 | <0.0001 |

Prevalence ratio (PR); CI: Confidence interval.

unadjusted estimate of 0.328 (95% CI: 0.285–0.378). After PSM adjustment, the prevalence of high school completion was 61.9% lower for those who had an early marriage (PR = 0.381; 95% CI: 0.298–0.488). Regression adjustment produced a PR estimate of 0.446 (95% CI: 0.374–0.532), indicating that the prevalence of high school completion was 55.4% lower for those who had an early marriage.

## Sensitivity analysis

Finally, we performed a sensitivity analysis (Table 3) to examine the magnitude of hidden bias that would alter our estimated treatment effects and inferences. In other words, we determined how robust our matching analysis is to unobserved confounding variables. The details of the sensitivity analysis procedure are given elsewhere [41].

Under the no hidden bias assumption ($\Gamma = 1$), the QMH test-statistic indicates a significant treatment effect. Considering the bounds that assume we have over-estimated the true treatment effect ($Q^+_{MH}$), the treatment effect is significant under $\Gamma = 1$. It becomes even more significant for increasing values of $\Gamma$. However, for the bounds under the assumption of under-estimating

**Table 3. Bias analysis of sensitivity to unmeasured confounding.**

| Gamma (Γ) | Q_mh+ | Q_mh- | p_mh+ | p_mh- |
|---|---|---|---|---|
| 1 | 10.685 | 10.685 | 0.000 | 0.000 |
| 1.1 | 11.445 | 9.946 | 0.000 | 0.000 |
| 1.2 | 12.143 | 9.274 | 0.000 | 0.000 |
| 1.3 | 12.794 | 8.661 | 0.000 | 0.000 |
| 1.4 | 13.404 | 8.099 | 0.000 | 0.000 |
| 1.5 | 13.980 | 7.580 | 0.000 | 0.000 |
| 1.6 | 14.526 | 7.096 | 0.000 | 0.000 |
| 1.7 | 15.045 | 6.645 | 0.000 | 0.000 |
| 1.8 | 15.541 | 6.221 | 0.000 | 0.000 |
| 1.9 | 16.015 | 5.822 | 0.000 | 0.000 |
| 2 | 16.471 | 5.445 | 0.000 | 0.000 |
| 2.1 | 16.909 | 5.087 | 0.000 | 0.000 |
| 2.2 | 17.332 | 4.747 | 0.000 | 0.000 |
| 2.3 | 17.741 | 4.422 | 0.000 | 0.000 |
| 2.4 | 18.136 | 4.113 | 0.000 | 0.000 |
| 2.5 | 18.520 | 3.816 | 0.000 | 0.000 |
| 2.6 | 18.893 | 3.532 | 0.000 | 0.000 |
| 2.7 | 19.255 | 3.258 | 0.000 | 0.001 |
| 2.8 | 19.608 | 2.995 | 0.000 | 0.001 |
| 2.9 | 19.952 | 2.742 | 0.000 | 0.003 |
| 3 | 20.288 | 2.497 | 0.000 | 0.006 |
| 3.1 | 20.617 | 2.261 | 0.000 | 0.012 |
| 3.2 | 20.937 | 2.032 | 0.000 | 0.021 |
| 3.3 | 21.252 | 1.811 | 0.000 | 0.035 |
| 3.4 | 21.559 | 1.596 | 0.000 | 0.055 |
| 3.5 | 21.861 | 1.387 | 0.000 | 0.083 |

Γ: odds of differential assignment due to unobserved factors. $Q^+_{MH}$: Mantel-Haenszel statistic (assumption: overestimation of treatment effect). $Q^-_{MH}$: Mantel-Haenszel statistic (assumption: underestimation of treatment effect). p-_mh: significance level (assumption: overestimation of treatment effect). p+_mh: significance level (assumption: underestimation of treatment effect).

the treatment effect ($Q^-_{MH}$), a value as high as $\Gamma = 3.4$ or more would be required for the results not to be significant at the 5% level. Therefore, our matching analysis suggests that early marriage is significantly associated with non-completion of high school education and is vulnerable to hidden bias. We should thus interpret the matching results with caution.

## Discussion and conclusions

The current study investigated the effect of child marriage on high school completion. This study suggests that early marriage is still prevalent in Zimbabwe and perhaps a hidden crisis given the limited attention in empirical research. We found that among women in the age group 20–29, approximately 37% were married before their 18th birthday, placing Zimbabwe amongst the countries with the highest rates of underage married girls in Africa [9, 10]. This enduring practice continues to undermine all efforts towards gender equality. It must, therefore, be urgently addressed if the country is to make progress towards meeting sustainable development goal number five.

The fact that child marriage is associated with low educational attainment is well known [23, 29, 49, 50]; however, the exact extent of the impact is less clear given relatively few studies have measured this aspect, especially in Zimbabwe. In this study, we used propensity score matching [43] to statistically balance women who were married early and those who married late in terms of various factors that could potentially influence their likelihood of completing the first cycle of high school. Consistent with the few existing quasi-experimental studies on early marriage [13, 30, 51], our results suggest women who were married before their 18[th] birthday had a lower propensity to complete the first cycle of high school relative to their peers with similar characteristics but had married after their 18[th] birthday. We also found evidence of confounding. Before matching, women who married as children were 55.4% (regression adjusted) less likely to complete lower secondary school. However, the co-efficient increased to 61.9% after matching, suggesting that self-selection plays a part in explaining the relationship.

What exactly do these findings mean for Zimbabwe? First, the results suggest that girl-child marriage is a significant barrier to educational attainment. If the practice of child marriage is not addressed, Zimbabwe will most likely fail to achieve sustainable development goals 4.2 and 5.3 which seek to i) ensure that all boys and girls complete primary and secondary education and ii) "eliminate all harmful practices, such as child, early and forced marriage" respectively [28]. Considering the risks associated with early marriage [15, 19, 32], delaying early marriage should be at the forefront of development policy in Zimbabwe. Clearly, legislation alone has failed to curb the practice [19, 25]; thus, there is a need for the Government of Zimbabwe and its local and global development partners to intensify existing campaigns against child marriage. In particular, social change interventions that target adults and counter beliefs about adolescent sexuality and prepubescent marriages should be put in place. Similar programmes have been implemented in countries such as India and Ethiopia, with considerable success in reducing both the prevalence of child marriage and altering social norms [52]. Second, given the empirically determined link between child marriage and educational attainment, education plans in the country should integrate the goal of eradicating child marriage. Interventions that target girls who drop out of school must also be put in place, while existing programmes that provide school subsidies for vulnerable young girls should be broadened [27]. Further, interventions that help keep young girls in school beyond the lower secondary level might be beneficial to eradicating child marriage. Being out of school has been linked to risky sexual behaviours and unintended pregnancies- both of which are catalysts for early marriage [52].

While propensity score matching in this study has enabled us to examine the effect of early marriage completion of lower secondary school, some limitations should be considered when

interpreting the findings. First, we acknowledge that the relationship between educational attainment and early marriage is complex as the latter can be both the cause and consequence of dropping out of school. Thus, there is a possibility of reverse causality between child marriage and educational attainment; however, PSM does not correct for this. Another limitation is that the DHS data does not include variables such as school quality or reasons for dropping out, which can shed more light on the temporal sequencing of child marriage and school leaving. In other words, more research is needed, which explores the effect of these factors on educational attainment so as to effectively tease out the effect of child marriage on secondary school completion. Finally, propensity score matching does not control for unobserved confounding. There may be other risk factors which we were not able to control.

Despite these limitations, our study makes significant contributions to the literature on child marriage in general and offers valuable insights into the phenomenon of child marriage in Zimbabwe. However, longitudinal and cohort studies are still needed to validate these findings as well as tease out causation.

## Acknowledgments

The authors would like to thank the DHS programme for making the data used in this study available.

## Author Contributions

**Conceptualization:** Annah V. Bengesai.

**Data curation:** Annah V. Bengesai.

**Formal analysis:** Annah V. Bengesai, Lateef B. Amusa.

**Methodology:** Annah V. Bengesai, Lateef B. Amusa.

**Writing – original draft:** Annah V. Bengesai, Lateef B. Amusa, Felix Makonye.

**Writing – review & editing:** Annah V. Bengesai, Lateef B. Amusa, Felix Makonye.

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
