## [Decision Letter · Decision Letter 0]

24 Nov 2020

PONE-D-20-28233

The impact of girl child marriage on the completion of  the first cycle of secondary education in Zimbabwe: a propensity score analysis

PLOS ONE

Dear Dr. Bengesai,

Thank you for submitting your manuscript to PLOS ONE. After careful consideration, we feel that it has merit but does not fully meet PLOS ONE’s publication criteria as it currently stands. Therefore, we invite you to submit a revised version of the manuscript that addresses the points raised during the review process.

As you will see, the reviewers have comments on a wide range of issues, the most important of which concern the appropriate of the methods used, the presentation of the results, and the discussion of the results. 

We look forward to receiving your revised manuscript.

Kind regards,

David Hotchkiss

Academic Editor

PLOS ONE

Journal Requirements:

Reviewers' comments:

Reviewer's Responses to Questions

**Comments to the Author**

1. Is the manuscript technically sound, and do the data support the conclusions?

Reviewer #1: No

Reviewer #2: Partly

2. Has the statistical analysis been performed appropriately and rigorously? 

Reviewer #1: No

Reviewer #2: No

3. Have the authors made all data underlying the findings in their manuscript fully available?

Reviewer #1: No

Reviewer #2: No

4. Is the manuscript presented in an intelligible fashion and written in standard English?

Reviewer #1: Yes

Reviewer #2: Yes

5. Review Comments to the Author

Reviewer #1: The paper highlights the importance of addressing child marriage by exploring its impact on secondary school completion in Zimbabwe. While the authors have done a nice job of reviewing the evidence base on child marriage in Zimbabwe, unfortunately the arguments are not sufficiently supported by rigorous evidence in a manner that the current analysis has data to address. The survey of literature presents interesting cultural factors such as sexual initiation, virgin pledging and pregnancy triggered marriages. However, the contributions of these factors are not reflected in the analysis. While it is understandable that marriage characteristics data is not available, it should be possible to explore the contribution of pregnancy preceding marriage. This is an important missed opportunity. Thus the review of the literature raises many more questions than it answers. Overall, at a substantive level, is an important considerations is to address whether child marriage really the critical driver of schooling or or are other factors at play such as early sexual initiation, unintended pregnancy, school quality and the absence of services to address these. Without considering these other factors, and acknowledging that child marriage itself may be driven by them, it is misleading to conclude that more needs to be done to child marriage, particularly as it is not clear what precisely can be recommended as a strategy for addressing child marriage directly rather than some of its other drivers. A second important reservation I have about the paper is in terms of methods. With the use of retrospective data, and without considering the temporal sequencing of school dropout and child marriage it is misleading to conclude that child marriage is the driver when the reverse may also be true. There are indeed several papers (cited below) that call attention to the fact that at least part of the association may be related to underlying factors that lead to school dropout, which then leads to early marriage and/or pregnancy. This possibility of reverse causality has important implications for policy. Solutions to improving school outcomes may well need to focus on school quality and not child marriage per se. Finally, the paper includes sample weights as a variable in propensity score matching. This is certainly unusual and it is not obvious to the current reviewer how it is justified. That needs to be explained better.

Biddlecom, Ann; Gregory, Richard; Lloyd, Cynthia B.; Mensch, Barbara S. Associations between premarital sex and leaving school in four Sub-Saharan African countries. Studies in Family Planning. 2008; 39(4):337–350. [PubMed: 19248719]

Birdthistle, Isolde; Floyd, Sian; Nyagadza, Auxillia; Mudziwapasi, Netsai; Gregson, Simon; Glynn, Judith R. Is education the link between orphanhood and HIV/HSV-2 risk among female adolescents in urban Zimbabwe? Social Science & Medicine. 2009; 68(10):1810–1818. [PubMed: 19303688]

Case, Anne; Ardington, Cally. The impact of parental death on school outcomes: Longitudinal evidence from South Africa. Demography. 2006; 43(3):401–420. [PubMed: 17051820]

Lloyd, Cynthia B.; Mensch, Barbara S. Marriage and childbirth as factors in dropping out from school: An analysis of DHS data from sub-Saharan Africa. Population Studies. 2008; 62(1):1–13. [PubMed:

Reviewer #2: The authors estimate the effect of child marriage on the probability of completing the ordinary level of secondary school in Zimbabwe. They recognize that this estimate is likely to be confounded and attempt to control for measured confounding using propensity score matching. I appreciate that the authors recognize the fact that estimates of this effect from observational studies are subject to a great deal of bias and likely far from the true causal effect. However, the analytic methods used in this study need to be better explained and justified before it is suitable for publication. My detailed comments on each section of the manuscript follow.

Introduction

The introduction and framing of this argument could be improved. In particular, the information on line 118 regarding results from Koski and Heymann (2018) is partially incorrect. That study found that immigrant children living in the United States were more likely to be married than their peers who were born in the United States, but immigrants comprise a minority of the population of the United States and so it is incorrect to state that the majority of those married were immigrants. Please correct this.

Lines 139-141 include the following sentence: “In some communities, such as in Iran and Nigeria, the onset of menarche is considered the threshold for adulthood. Hence, girls who reach this biological threshold are perceived to be ready for marriage. (14, 17).” It is unlikely that all citizens of these two nations hold this belief and the authors should be more cautious in their interpretation. Reference number 17 is not sufficient to support this statement in Nigeria. It appears to include a single sentence that is not based on research and, unfortunately, it is published in a known predatory outlet, making it unclear whether the work was peer reviewed.

Data and statistical analyses

The paragraph that explains the educational structure in Zimbabwe (lines 260-271) is important. However, the authors do not address how well this structure corresponds with the measured variables available in DHS data, which typically include a continuous measure of the total number of years of schooling and a second categorical measure that indicates the highest level (no schooling, primary, secondary, or more than secondary) an individual attended or completed. (Notably, the 2015 DHS report for Zimbabwe includes the categories listed above and does not differentiate between completion of the ordinary and advanced levels of secondary school.) How did the authors use these variables to capture completion of the ordinary level of secondary school versus the advanced level? What is the expected magnitude of misclassification of educational attainment, and what effect would such misclassification be expected to have on their estimates?

The authors appear to confuse the concept of selection bias (line 308, line 439) with confounding, or may be using these terms interchangeably, which is confusing and should be corrected. (In epidemiology, all of the bias discussed in this paper would be categorized as confounding.) I agree with their assertion that estimates of the association between child marriage and educational attainment are very seriously confounded and could result from reverse causality. Hypothetically speaking, in order to identify the causal effect of child marriage on this outcome, one would need to compare a group of girls who married prior to the age of 18 with a group who married at the age of 18 years or later and those girls would need to be exactly the same in every way except for their age at the time of their first marriage. Clearly, it is very difficult to attain such a comparison in an observational study based on DHS data. In reality, girls who marry before the age of 18 are very different from those who marry at later ages in many ways. Unfortunately, the most important confounders of this relationship, such as childhood socioeconomic conditions and attitudes toward gender equality among the girls’ family and community prior to her marriage, are not captured in the DHS. This severely limits the extent to which confounding can be controlled, even through use of propensity score matching.

The authors use propensity score matching in an effort to make the treatment groups more exchangeable with regard to a small number of measured variables, but the rationale for the inclusion and treatment of these variables and the omission of others is unclear. For example, it is unclear why the respondent’s age and the age difference between spouses were categorized. Why not use age and age difference as continuous variables to improve the granularity of matching (and therefore further limit confounding)? Also concerning is that any couples in which the wife was older than her husband would yield a negative number, and negative numbers of any magnitude appear to be grouped in the category �0-4 years. Are the authors sure that all women in their sample were in their first marriage? If not, the age difference between the woman and her second (or later) partner may not be relevant to this analysis. Why was ethnicity not included in the propensity score model? It is one of the few pre-exposure variables available to the authors and is strongly correlated with child marriage in many countries, though I am not familiar with how this is distributed in Zimbabwe.

The authors cite work by Donald Rubin, a pioneer in the development of propensity score methods, but in some cases do so erroneously. For example, Rubin does not advocate for throwing any and all variables into the propensity score model as suggested on lines 320-321; in particular, including variables that may be a consequence of the exposure (in this case, marriage before the age of 18) is typically inappropriate. This is where use of cross-sectional surveys such as the DHS becomes even more problematic. Most variables in the DHS are measured in adulthood, well after marriage, and may be influenced by that marriage. For example, the categorical measure of household-level socioeconomic status included in all DHS is based on characteristics at the time the survey was conducted. One can easily imagine that child marriage may be a means for parents to secure places for their daughters in households with a higher standard of living; the authors acknowledge this in the introduction to their paper. This makes it inappropriate to adjust for this variable. The same argument could be made for place of residence measured after marriage.

The specific determination of matches based on propensity scores is also insufficiently explained. For example, the authors state that they used nearest neighbor matching, but more information is needed. Were calipers used to define a maximum acceptable difference in scores to assign a match? Were controls sampled with replacement? How many observations were off support, meaning that no reasonable match could be found?

The authors include estimates of the absolute standardized differences in their results. Why are values for these differences missing for some values in Table 1?

The authors refer to estimates of “heterogeneity” in the treatment effect (line 237, lines 470-472), but no such analyses were conducted as far as I can tell. The estimates in Table 2 appear to be for the entire sample and no discussion of tests for heterogeneity across sup-groups is included.

Given that data from this survey is cross-sectional and measures prevalence, I recommend that the authors report prevalence differences and prevalence ratios rather than odds ratios. Including absolute measures of effect (i.e. prevalence differences) would strengthen the paper substantially. This can be done after running a logistic regression model by using the – margins – command in Stata.

Discussion

The term “quasi-experimental” (line 432) typically refers to natural experiments in which exposure is plausibly random; it is inappropriate in this study in which exposure is far from random and a large degree of unmeasured and residual confounding likely remain in the estimates.

Lines 473-475 seem to indicate a fundamental misunderstanding of the DHS. The survey collects data from a nationally representative sample of individuals; the data are at the individual-level, not at the national-level.

I find the authors’ discussion of the limitations of their research cursory and insufficient. For example, “…our data was drawn from self-reported responses; thus, there is a possibility of underreporting of sensitive information.” What information used in this study would they categorize as sensitive and why? Would such underreporting be likely to bias their estimates? They note that propensity score matching does not control for unmeasured confounding but say nothing about the very high likelihood that their results remain substantially confounded even after accounting for the variables included in their propensity score model. Quantitative bias analysis to estimate how robust their estimates are to the presence of unmeasured and residual confounding would improve the paper immensely.

6. PLOS authors have the option to publish the peer review history of their article (what does this mean?). If published, this will include your full peer review and any attached files.

Reviewer #1: **Yes: **Sajeda Amin

Reviewer #2: **Yes: **Alissa Koski

---

## [Author Response · Author response to Decision Letter 0]

29 Jan 2021

Response to Reviewers

Dear Editor,

We would like to thank the reviewers for their comments as well as the opportunity to revise our manuscript for further consideration in your journal. We have addressed all the comments from the reviewers and below we provide a detailed description of the revisions made. These changes are also highlighted in yellow in the revised manuscript we submitted.

Reviewer #1

 The paper highlights the importance of addressing child marriage by exploring its impact on secondary school completion in Zimbabwe. While the authors have done a nice job of reviewing the evidence base on child marriage in Zimbabwe, unfortunately the arguments are not sufficiently supported by rigorous evidence in a manner that the current analysis has data to address. The survey of literature presents interesting cultural factors such as sexual initiation, virgin pledging and pregnancy triggered marriages. However, the contributions of these factors are not reflected in the analysis. While it is understandable that marriage characteristics data is not available, it should be possible to explore the contribution of pregnancy preceding marriage. This is an important missed opportunity. Thus the review of the literature raises many more questions than it answers. Overall, at a substantive level, is an important consideration is to address whether child marriage really the critical driver of schooling or or are other factors at play such as early sexual initiation, unintended pregnancy, school quality and the absence of services to address these. Without considering these other factors, and acknowledging that child marriage itself may be driven by them, it is misleading to conclude that more needs to be done to child marriage, particularly as it is not clear what precisely can be recommended as a strategy for addressing child marriage directly rather than some of its other drivers.

Response: We concur that early sexual debut and teenage pregnancy are reciprocal to early marriage as well as educational attainment. Therefore, we included age at sexual debut as well as age at first birth as controls. 

A second important reservation I have about the paper is in terms of methods. With the use of retrospective data, and without considering the temporal sequencing of school dropout and child marriage it is misleading to conclude that child marriage is the driver when the reverse may also be true. There are indeed several papers (cited below) that call attention to the fact that at least part of the association may be related to underlying factors that lead to school dropout, which then leads to early marriage and/or pregnancy.

Response: The DHS does not include any variables that measure school type or quality. We acknowledge this as a limitation in our study. 

 This possibility of reverse causality has important implications for policy. Solutions to improving school outcomes may well need to focus on school quality and not child marriage per se. 

Response: While we acknowledge the relevance of school quality on educational attainment, we cannot make conclusions on this as school quality was not measured in the ZDHS. We acknowledge this as a limitation. 

Finally, the paper includes sample weights as a variable in propensity score matching. This is certainly unusual and it is not obvious to the current reviewer how it is justified. That needs to be explained better.

Response: Though it is correct to say that it's not a usual approach, including survey weights in the propensity score model as an additional covariate is one of the recommended methods for estimating population-level treatment effects (see references 1,2 below). However, for simplicity sake, we have modified the propensity score matching analyses to accommodate survey weights when estimating the effect of treatment, but not when estimating the propensity score model (as suggested by reference 3 below). 

Austin PC, Jembere N, Chiu M. Propensity score matching and complex surveys. Statistical methods in medical research. 2018 Apr;27(4):1240-57.

DuGoff EH, Schuler M, Stuart, EA. Generalising observational study results: applying propensity score methods to complex surveys. Health Serv Res. 2014; 49(1): 284-303. DOI: 10.1111/1475-6773.12090 

Zanutto EL. A comparison of propensity score and linear regression analysis of complex survey data. Journal of data Science. 2006 Jan 1;4(1):67-91.

Reviewer #2: The authors estimate the effect of child marriage on the probability of completing the ordinary level of secondary school in Zimbabwe. They recognize that this estimate is likely to be confounded and attempt to control for measured confounding using propensity score matching. I appreciate that the authors recognize the fact that estimates of this effect from observational studies are subject to a great deal of bias and likely far from the true causal effect. However, the analytic methods used in this study need to be better explained and justified before it is suitable for publication. My detailed comments on each section of the manuscript follow.

Introduction

The introduction and framing of this argument could be improved. In particular, the information on line 118 regarding results from Koski and Heymann (2018) is partially incorrect. That study found that immigrant children living in the United States were more likely to be married than their peers who were born in the United States, but immigrants comprise a minority of the population of the United States and so it is incorrect to state that the majority of those married were immigrants. Please correct this.

Response: Corrected as suggested.

Lines 139-141 include the following sentence: “In some communities, such as in Iran and Nigeria, the onset of menarche is considered the threshold for adulthood. Hence, girls who reach this biological threshold are perceived to be ready for marriage. (14, 17).” It is unlikely that all citizens of these two nations hold this belief and the authors should be more cautious in their interpretation. Reference number 17 is not sufficient to support this statement in Nigeria. It appears to include a single sentence that is not based on research and, unfortunately, it is published in a known predatory outlet, making it unclear whether the work was peer reviewed.

Response: Changed to: In some communities, the onset of menarche is considered the threshold for adulthood and sign of readiness for marriage.

Reference also changed to: Raj A, Ghule M, Nair S, Saggurti N, Balaiah D, Silverman JG. Age at menarche, education, and child marriage among young wives in rural Maharashtra, India. Int J Gynaecol Obstet. 2015

Data and statistical analyses

The paragraph that explains the educational structure in Zimbabwe (lines 260-271) is important. However, the authors do not address how well this structure corresponds with the measured variables available in DHS data, which typically include a continuous measure of the total number of years of schooling and a second categorical measure that indicates the highest level (no schooling, primary, secondary, or more than secondary) an individual attended or completed. (Notably, the 2015 DHS report for Zimbabwe includes the categories listed above and does not differentiate between completion of the ordinary and advanced levels of secondary school.) How did the authors use these variables to capture completion of the ordinary level of secondary school versus the advanced level? What is the expected magnitude of misclassification of educational attainment, and what effect would such misclassification be expected to have on their estimates?

Response: We used information on years of schooling and deduced that those who completed 11 years of schooling would have completed the Ordinary level of Education, based on the 7-4-2 education structure in Zimbabwe. 

The authors appear to confuse the concept of selection bias (line 308, line 439) with confounding, or may be using these terms interchangeably, which is confusing and should be corrected. (In epidemiology, all of the bias discussed in this paper would be categorized as confounding.) I agree with their assertion that estimates of the association between child marriage and educational attainment are very seriously confounded and could result from reverse causality. Hypothetically speaking, in order to identify the causal effect of child marriage on this outcome, one would need to compare a group of girls who married prior to the age of 18 with a group who married at the age of 18 years or later and those girls would need to be exactly the same in every way except for their age at the time of their first marriage. Clearly, it is very difficult to attain such a comparison in an observational study based on DHS data. In reality, girls who marry before the age of 18 are very different from those who marry at later ages in many ways. Unfortunately, the most important confounders of this relationship, such as childhood socioeconomic conditions and attitudes toward gender equality among the girls’ family and community prior to her marriage, are not captured in the DHS. This severely limits the extent to which confounding can be controlled, even through use of propensity score matching.

Response: Replace selection bias with confounding

Response: done as suggested

The authors use propensity score matching in an effort to make the treatment groups more exchangeable with regard to a small number of measured variables, but the rationale for the inclusion and treatment of these variables and the omission of others is unclear. For example, it is unclear why the respondent’s age and the age difference between spouses were categorized. Why not use age and age difference as continuous variables to improve the granularity of matching (and therefore further limit confounding)? Also concerning is that any couples in which the wife was older than her husband would yield a negative number, and negative numbers of any magnitude appear to be grouped in the category �0-4 years. 

Response: As suggested, we have used age as a continuous variable. However, we chose to leave age difference as categorical to make it easier to use the missing indicator method to deal with its missing values, which are quite many.

Are the authors sure that all women in their sample were in their first marriage? If not, the age difference between the woman and her second (or later) partner may not be relevant to this analysis.

Response: We included the variable union, to control for whether the woman was in the first or subsequent union.

 Why was ethnicity not included in the propensity score model? It is one of the few pre-exposure variables available to the authors and is strongly correlated with child marriage in many countries, though I am not familiar with how this is distributed in Zimbabwe.

Response: We did not include ethnicity as this was not measured in the 2015 ZDHS

The authors cite work by Donald Rubin, a pioneer in the development of propensity score methods, but in some cases do so erroneously. For example, Rubin does not advocate for throwing any and all variables into the propensity score model as suggested on lines 320-321; in particular, including variables that may be a consequence of the exposure (in this case, marriage before the age of 18) is typically inappropriate. This is where use of cross-sectional surveys such as the DHS becomes even more problematic. Most variables in the DHS are measured in adulthood, well after marriage, and may be influenced by that marriage. For example, the categorical measure of household-level socioeconomic status included in all DHS is based on characteristics at the time the survey was conducted. One can easily imagine that child marriage may be a means for parents to secure places for their daughters in households with a higher standard of living; the authors acknowledge this in the introduction to their paper. This makes it inappropriate to adjust for this variable. The same argument could be made for place of residence measured after marriage.

Response: We have only included variables that affect either the exposure or outcome, not those that may be a consequence of the exposure. 

We also acknowledge the possibility of measurement error, given the ZDHS only provides current household wealth information, and not prior to the marriage. However, it is not unreasonable to assume that young women are likely to marry within the same socio-economic status (12). As well, it is plausible that those who marry in wealthy families might be allowed to continue with their education post marriage. Taking this view, we contend that estimations which include household wealth are superior to those without, as they give us some indication of the effect of socio-economic status, albeit with certain limitations. 

The specific determination of matches based on propensity scores is also insufficiently explained. For example, the authors state that they used nearest neighbor matching, but more information is needed. Were calipers used to define a maximum acceptable difference in scores to assign a match? Were controls sampled with replacement? How many observations were off support, meaning that no reasonable match could be found?

Response: Done as suggested. More details about the propensity score matching have been included.

The authors include estimates of the absolute standardized differences in their results. Why are values for these differences missing for some values in Table 1?

Response: the values missing have now been included appropriately 

The authors refer to estimates of “heterogeneity” in the treatment effect (line 237, lines 470-472), but no such analyses were conducted as far as I can tell. The estimates in Table 2 appear to be for the entire sample and no discussion of tests for heterogeneity across sup-groups is included.

Response: references to heterogeneity removed 

Given that data from this survey is cross-sectional and measures prevalence, I recommend that the authors report prevalence differences and prevalence ratios rather than odds ratios. Including absolute measures of effect (i.e. prevalence differences) would strengthen the paper substantially. This can be done after running a logistic regression model by using the – margins – command in Stata.

Response: done as suggested

Discussion

The term “quasi-experimental” (line 432) typically refers to natural experiments in which exposure is plausibly random; it is inappropriate in this study in which exposure is far from random and a large degree of unmeasured and residual confounding likely remain in the estimates.

Response: Corrected as suggested. 

Lines 473-475 seem to indicate a fundamental misunderstanding of the DHS. The survey collects data from a nationally representative sample of individuals; the data are at the individual-level, not at the national-level.

Response: This limitation has been removed.

I find the authors’ discussion of the limitations of their research cursory and insufficient. For example, “…our data was drawn from self-reported responses; thus, there is a possibility of underreporting of sensitive information.” What information used in this study would they categorize as sensitive and why? Would such underreporting be likely to bias their estimates? They note that propensity score matching does not control for unmeasured confounding but say nothing about the very high likelihood that their results remain substantially confounded even after accounting for the variables included in their propensity score model. Quantitative bias analysis to estimate how robust their estimates are to the presence of unmeasured and residual confounding would improve the paper immensely.

Response: The limitations have been revised as follows: 

While the use of propensity score matching in this study has enabled us to examine the effect of early marriage completion of lower secondary school, some limitations should be considered when interpreting the findings. First, we acknowledge that the relationship between educational attainment and early marriage is not straightforward as the latter can be both the cause and consequence of dropping out of school Thus, there is a possibility of reverse causality between child marriage and educational attainment; however, PSM does not correct for this. Another limitation is that the DHS data does not include variables such as quality of school or reasons for dropping out, which can shed more light on the temporal sequencing of child marriage and school leaving. More research is needed which explores the effect of these factors on educational attainment so as to effectively tease out the effect of child marriage on secondary school completion. Finally, propensity score matching does not control for unobserved confounding. There may be other risk factors which we were not able to control.

Sincerely,

Dr Annah Bengesai

University of KwaZulu-Natal

South Africa

Email: bengesai@ukzn.ac.za

ORCID 0000-0002-2711-8530

Dr Lateef A. Babatunde 

University of Ilorin, Nigeria

Email: Amusa.lb@unilorin.edu.ng

And 

Dr Felix Makhonye

University of KwaZulu-Natal

Email: MakonyeF@ukzn.ac.za

---

## [Decision Letter · Decision Letter 1]

18 Feb 2021

PONE-D-20-28233R1

The impact of girl child marriage on the completion of the first cycle of secondary education in Zimbabwe: a propensity score analysis

PLOS ONE

Dear Dr. Bengesai,

Thank you for submitting your manuscript to PLOS ONE. After careful consideration, we feel that it has merit but does not fully meet PLOS ONE’s publication criteria as it currently stands. Therefore, we invite you to submit a revised version of the manuscript that addresses the points raised during the review process.

As you will see, one of the reviewers has a number of comments on how you addressed the comments on the initial version of the manuscript.  Please address these comments an provide point-by-point responses. 

We look forward to receiving your revised manuscript.

Kind regards,

David Hotchkiss

Academic Editor

PLOS ONE

Reviewers' comments:

Reviewer's Responses to Questions

**Comments to the Author**

1. If the authors have adequately addressed your comments raised in a previous round of review and you feel that this manuscript is now acceptable for publication, you may indicate that here to bypass the “Comments to the Author” section, enter your conflict of interest statement in the “Confidential to Editor” section, and submit your "Accept" recommendation.

Reviewer #2: (No Response)

2. Is the manuscript technically sound, and do the data support the conclusions?

Reviewer #2: No

3. Has the statistical analysis been performed appropriately and rigorously? 

Reviewer #2: No

4. Have the authors made all data underlying the findings in their manuscript fully available?

Reviewer #2: Yes

5. Is the manuscript presented in an intelligible fashion and written in standard English?

Reviewer #2: Yes

6. Review Comments to the Author

Reviewer #2: The revised manuscript is much improved with regard to the description of how the outcome variable was measured and the inclusion of a sensitivity analysis. However, the fundamental problems with confounder identification and adjustment remain and must be addressed.

In my initial review I recommended that the authors measure age and age difference variables continuously to improve the accuracy of matching. This was done for the wife’s age but not for the age difference between spouses. I don’t follow the logic of using spousal age difference a categorical variable because there were a large proportion of missing values. Missing values for wife’s age and/or husband’s age would prohibit the estimation of age difference regardless of whether the variable was measured continuously or categorically. Moreover, it is unclear how missing values were handled in the analysis. At a later point in the paper (lines 371-374) the authors state, “The missing indicator method was applied to missing values on covariates such as spousal age difference…” but it is unclear what this means and no citations are provided. If this refers to the use of binary variables to indicate missingness, the authors should acknowledge the high likelihood that this method itself induces bias. (For reference, see Donders et al. Journal of Clinical Epidemiology 2006.)

In my initial review I raised concerns regarding control for variables that may be affected by the exposure, including household socioeconomic status (SES). The authors assert that it is reasonable to assume that young women marry husbands of the same SES, suggesting that SES is the same prior to and after marriage. This contradicts much of what is known about the relationship between poverty and child marriage. It is also contradicted by the introduction to the paper, which indicates that “…kuzvarira is often a survival tactic where low-income families negotiate with wealthy families to marry off their daughters at a younger age in exchange for grains, cows or money.” This certainly suggests that SES differs prior to and after marriage, especially in the case of child marriage, and should not be adjusted for. For further information on identifying potential confounders, I recommend that the authors consult recent work by VanderWeele (Principles of confounder selection, European Journal of Epidemiology, 2019).

In my initial review I asked for clarification regarding whether all women were in their first unions. Given that the DHS only collects information on a woman’s current spouse, women in second or later unions would not have provided relevant information about their first spouse. In response, the authors included a variable for union number in the model. Union number cannot possibly affect age at marriage and therefore cannot be a confounder. It should not be adjusted for. An alternative would be to exclude women who were not in their first unions from the analysis, though this may result in problems with selection bias.

In the revised version of the manuscript the authors have added age at first sex and age at first birth to their model. While sex and pregnancy may lead to child marriage, for a substantial number of young girls, age at marriage probably determines age at first sex and age at first birth. For girls whose first birth follows their marriage, this variable is almost certainly on the causal pathway between age at marriage and educational attainment. Again, this strongly indicates that these variables should not be treated as confounders. The proportion of girls who report an age at first sex prior to age at first marriage (which can be estimated using DHS data) might guide subsequent analytic decisions.

On line 463 the authors report the prevalence of child marriage among women between 17 and 24 years of age. This doesn’t correspond with the analytic sample of 20-29-year-olds described in the rest of the paper. Is this a typing error?

7. PLOS authors have the option to publish the peer review history of their article (what does this mean?). If published, this will include your full peer review and any attached files.

Reviewer #2: **Yes: **Alissa Koski

---

## [Author Response · Author response to Decision Letter 1]

5 Apr 2021

Dear Editor,

We would like to thank the reviewers for their comments as well as the opportunity to revise our manuscript for further consideration in your journal. We have addressed all the comments from the reviewers and below we provide a detailed description of the revisions made. These changes are also highlighted in yellow in the revised manuscript we submitted.

Reviewer #2: The revised manuscript is much improved with regard to the description of how the outcome variable was measured and the inclusion of a sensitivity analysis. However, the fundamental problems with confounder identification and adjustment remain and must be addressed.

In my initial review I recommended that the authors measure age and age difference variables continuously to improve the accuracy of matching. This was done for the wife’s age but not for the age difference between spouses. I don’t follow the logic of using spousal age difference a categorical variable because there were a large proportion of missing values. Missing values for wife’s age and/or husband’s age would prohibit the estimation of age difference regardless of whether the variable was measured continuously or categorically. Moreover, it is unclear how missing values were handled in the analysis. At a later point in the paper (lines 371-374) the authors state, “The missing indicator method was applied to missing values on covariates such as spousal age difference…” but it is unclear what this means and no citations are provided. If this refers to the use of binary variables to indicate missingness, the authors should acknowledge the high likelihood that this method itself induces bias. (For reference, see Donders et al. Journal of Clinical Epidemiology 2006.)

Response: we have used both age and spousal age difference as continuous variables 

In my initial review I raised concerns regarding control for variables that may be affected by the exposure, including household socioeconomic status (SES). The authors assert that it is reasonable to assume that young women marry husbands of the same SES, suggesting that SES is the same prior to and after marriage. This contradicts much of what is known about the relationship between poverty and child marriage. It is also contradicted by the introduction to the paper, which indicates that “…kuzvarira is often a survival tactic where low-income families negotiate with wealthy families to marry off their daughters at a younger age in exchange for grains, cows or money.” This certainly suggests that SES differs prior to and after marriage, especially in the case of child marriage, and should not be adjusted for. For further information on identifying potential confounders, I recommend that the authors consult recent work by VanderWeele (Principles of confounder selection, European Journal of Epidemiology, 2019).

Response: we have removed wealth quintile from the analysis. 

In my initial review I asked for clarification regarding whether all women were in their first unions. Given that the DHS only collects information on a woman’s current spouse, women in second or later unions would not have provided relevant information about their first spouse. In response, the authors included a variable for union number in the model. Union number cannot possibly affect age at marriage and therefore cannot be a confounder. It should not be adjusted for. An alternative would be to exclude women who were not in their first unions from the analysis, though this may result in problems with selection bias.

Response: we have excluded women who were not in the first union from the analysis

In the revised version of the manuscript the authors have added age at first sex and age at first birth to their model. While sex and pregnancy may lead to child marriage, for a substantial number of young girls, age at marriage probably determines age at first sex and age at first birth. For girls whose first birth follows their marriage, this variable is almost certainly on the causal pathway between age at marriage and educational attainment. Again, this strongly indicates that these variables should not be treated as confounders. The proportion of girls who report an age at first sex prior to age at first marriage (which can be estimated using DHS data) might guide subsequent analytic decisions.

We have differentiated between women who had sexual debut before marriage and those whose debut was after marriage

On line 463 the authors report the prevalence of child marriage among women between 17 and 24 years of age. This doesn’t correspond with the analytic sample of 20-29-year-olds described in the rest of the paper. Is this a typing error?

We have corrected the error

We have also copy edited the manuscript for grammatical errors.

---

## [Decision Letter · Decision Letter 2]

17 May 2021

The impact of girl child marriage on the completion of the first cycle of secondary education in Zimbabwe: a propensity score analysis

PONE-D-20-28233R2

Dear Dr. Bengesai,

We’re pleased to inform you that your manuscript has been judged scientifically suitable for publication and will be formally accepted for publication once it meets all outstanding technical requirements.

Kind regards,

David Hotchkiss

Academic Editor

PLOS ONE

Additional Editor Comments (optional):

Reviewers' comments:

Reviewer's Responses to Questions

**Comments to the Author**

1. If the authors have adequately addressed your comments raised in a previous round of review and you feel that this manuscript is now acceptable for publication, you may indicate that here to bypass the “Comments to the Author” section, enter your conflict of interest statement in the “Confidential to Editor” section, and submit your "Accept" recommendation.

Reviewer #2: All comments have been addressed

2. Is the manuscript technically sound, and do the data support the conclusions?

Reviewer #2: Yes

3. Has the statistical analysis been performed appropriately and rigorously? 

Reviewer #2: Yes

4. Have the authors made all data underlying the findings in their manuscript fully available?

Reviewer #2: Yes

5. Is the manuscript presented in an intelligible fashion and written in standard English?

Reviewer #2: Yes

6. Review Comments to the Author

Reviewer #2: (No Response)

7. PLOS authors have the option to publish the peer review history of their article (what does this mean?). If published, this will include your full peer review and any attached files.

Reviewer #2: **Yes: **Alissa Koski

---

## [Editor Report · Acceptance letter]

21 May 2021

PONE-D-20-28233R2 

The impact of girl child marriage on the completion of the first cycle of secondary education in Zimbabwe: a propensity score analysis 

Dear Dr. Bengesai:

I'm pleased to inform you that your manuscript has been deemed suitable for publication in PLOS ONE. Congratulations! Your manuscript is now with our production department. 

Kind regards, 

on behalf of

Dr. David Hotchkiss 

Academic Editor

PLOS ONE